# Evaluation of Carbon Dioxide Laser–Assisted Treatment for Gingival Melanin Hyperpigmentation

**DOI:** 10.3390/dj10120238

**Published:** 2022-12-13

**Authors:** Trung Huynh Tran, Quynh Le Diem Nguyen, Thao Thi Do, Khue Nhut Truong, Quang Vinh Dang, Man Thi Ngoc Bui

**Affiliations:** 1Department of Oral Pathology and Periodontology, Can Tho University of Medicine and Pharmacy, Can Tho City 900000, Vietnam; 2Faculty of Odonto–Stomatology, Can Tho University of Medicine and Pharmacy, Can Tho City 900000, Vietnam; 3Department of Oral and Maxillofacial Surgery, Can Tho University of Medicine and Pharmacy, Can Tho City 900000, Vietnam

**Keywords:** gingival hyperpigmentation, melanin, CO_2_ laser, DOPI

## Abstract

Background: Smile aesthetics has a vital role to play in an individual’s life and one of the factors affecting the beauty of the smile is gingival color. A gingival color change or gingival hyperpigmentation causes an unesthetic smile line, especially in patients with a gummy smile, which is also known as a black gummy smile. Numerous gingival depigmentation methods have been performed successfully for ablating gingival melanin pigmented epithelium. Thus, the aim of this study is to evaluate the treatment efficacy of gingival hyperpigmentation by using a carbon dioxide (CO_2_) laser. Methods: A cross-sectional descriptive study was carried out with 38 patients at a hospital in Vietnam. Ponnaiyan classification and the Hedin melanin index were used to assess the distribution and extent of gingival pigmentation in the study. Pain assessment was performed using the Visual Analog Scale (VAS) to evaluate the intensity of pain during the laser treatment. In addition, clinical evaluation (i.e., wound healing) of each treatment procedure was conducted using the three level Dummett–Gupta Oral Pigmentation Index (DOPI) assessment. Results: This study showed that less pain was experienced by patients treated by CO_2_ laser; the rates of no pain, mild pain and moderate pain after treatment were, respectively, 21%, 76% and 2.6%; there was 100% complete epithelization after 1 week. The DOPI rates for turning from a DOPI score of 1, 2 or 3 to a DOPI score of 0 after a 12-week treatment were 87.5%, 76.9% and 24%, respectively. Conclusions: Using a CO_2_ laser for gingival melanin pigmentation treatment is a safe and effective procedure.

## 1. Introduction

Smile aesthetics are a significant part of a person’s daily life, and are influenced by various factors such as the teeth, lips and, especially, gingival color [1]. A smile is determined conventionally attractive by rosy or pink gingival tissue; thereby, any uncommon discoloration of the gingiva affects general facial expressions. Gingival pigmentation causes an unesthetic smile line. Gingival hyperpigmentation in an uncovered smile, also known as a black gummy smile, is a frequent concern in patients with a gingival smile, resulting in a lack of confidence in communication. Gum pigmentation is caused by melanin, melanoid, oxyhemoglobin, reduced hemoglobin and carotene [1]. Therefore, the overproduction of melanin, an endogenous pigment, is the common cause of gingival hyperpigmentation.

Melanin pigmentation is prevalently found in African and East Asian populations. Gingival hyperpigmentation is not just localized in a specific geographic area but can occur in any populations [2,3]. Some intrinsic factors such as skin pigment, smoking behaviour and heredity have also been found to be associated with gingival hyperpigmentation [4,5,6].

Gingival depigmentation is a treatment for removing melanin hyperpigmentation from gingiva. Several methods have received recognition from many authors including the use of a scalpel [7], electrosurgery [8] and gingival resection, each with their specific advantages. However, these treatments are invasive and excessive ablating methods that can cause patients discomfort and clinicians difficulties. Therefore, lasers have been developed and become potential alternatives. In the process of developing the modern laser technique, many kinds of laser treatments have been performed in dentistry, such as diode [8,9], Nd:YAG [10,11], Er,Cr:YSGG [12], Er:YAG [13,14,15] and CO_2_ laser [13,15,16]. They have provided some significant beneficial effects of the elimination of gingival melanin pigmentation that showed that the limitations of the past techniques could be remarkably reduced. Noninvasive, bloodless, painless and time-saving procedures have shown that most patients’ experience has been improved and dental practicians can achieve satisfactory results [17,18].

Due to different wavelengths, lasers have been employed for multi-purposes, namely, gingivectomy, gingivoplasty, frenectomy, incisional and excisional biopsy, soft tissue tuberosity reduction, operculectomy, coagulation of donor sites and soft tissue augmentation around osseointegrated implants [19,20,21]. In particular, the CO_2_ laser has the advantage of hemostasis with a shallow depth of penetration into soft tissue [22,23]. It also has the highest absorbance compared to other lasers and is absorbed by water, which accounts for 70% of biological tissues [18,23]. Hence, CO_2_ laser treatment is often performed in soft tissue surgery, set at 5–15 watts with either a pulsed or continuous mode [16]. This study was performed to evaluate the effects of the CO_2_ laser applied for gingival depigmentation.

## 2. Materials and Methods

### 2.1. Subjects

This cross-sectional descriptive study was conducted from March 2020 to January 2021. Thirty-eight patients, who complained of black gums and requested cosmetic correction, were selected from the outpatient section of Faculty of Odonto-Stomatology, Can Tho University of Medicine and Pharmacy Hospital, Vietnam.

In the study, 10 maxillary teeth of each patient from tooth 15–25, exhibiting moderate to severe melanin pigmentation in the maxillary anterior gingiva region from the right canine to the left canine (aesthetic zone), were selected. Patients with habit of smoking or abuse of any other oral substances, gum recession, missing teeth in the anterior region, patients on any forms of Non-Steroidal Anti-Inflammatory Drugs (NSAIDs), pregnant women, lactating mothers and patients with any systemic conditions were excluded from the study.

### 2.2. Study Methods

Initially, patients were given a topical spray anesthesia in the operating area and, if prolonged pain, were given a second anesthesia in the form of injection during the procedure.

CO_2_ laser (Millennium Ultra UP, KMI, Seoul, Republic of Korea) was used with 10,600 nm wavelength, 2.5 W power, 20 Hz frequency with a pulse duration 160 ms, energy density 79.5 J/cm^2^ per pulse, 1590 J/cm^2^ per second, continuous emission, with noncontact mode and beam diameter of 0.8 mm. Melanin pigmented gingivae were ablated by CO_2_ laser vaporization. Adjacent teeth were protected from the laser beam by applying an acrylic template to cover the labial of teeth. The procedure was repeated until ooze of small amount of blood was found, which meant that the laser had eliminated the epithelium, whereas leaving connective tissue layer and the desired depth of gingival tissue removal was attained. Preoperative and postoperative images were taken using a digital camera (Canon EOS 70D, Tokyo, Japan) in 50 mm focal length, aperture f11, ISO 100 with ring flash. The photographs were taken by the camera with the same focal length, light intensity, and the distance from the camera to the subject of 30 cm.

After completion of each procedure, the operated area was finally cleaned with gauze soaked with normal saline, and no dressing was given in any of the treated sites. The patient was instructed to avoid spicy, hard, sour and hot food, avoid smoking and brushing on the treated area, maintain oral hygiene by regular rinsing after meals, and advised to use warm saline rinses in the next days. The postoperative evaluation was carried out on the 1st, 7th day, and subsequently after 21 days from the depigmentation treatment about gingival color, gingival healing, pain intensity, bleeding and postoperative patient satisfaction.

### 2.3. Study Paremeters

In this study, the parameters used were:

The classification of gingival pigmentation based on anatomic delineation was presented with 6 classes by Deepa Ponnaiyan in 2013 [5], including:

Class I: only in attached gingiva.

Class II: both attached gingiva and interdental gingiva.

Class III: all parts (attached gingiva, marginal gingiva, and interdental gingiva).

Class IV: only in marginal gingiva.

Class V: only in interdental gingiva.

Class VI: both marginal gingiva and interdental gingiva.

Hedin’s melanin index classified gingival pigmentation [4], as follows:

Degree 1: one or two solitary unit(s) of pigmentation without the formation of a continuous ribbon.

Degree 2: more than three units of pigmentation without the formation of continuous ribbon.

Degree 3: one or more short continuous ribbons of pigmentation.

Degree 4: long continuous ribbon.

Pain assessment was performed using Visual Analog Scale (VAS) to measure intensity of pain experienced after 1 week, 4 weeks, 12 weeks of treatment [13]. Pain intensity was assessed on the 0–10 scale, with:

0: no pain.

0.1–3: mild pain.

3.1–6: moderate pain.

6.1–10: severe pain.

The clinical evaluation of gingival wound healing was used after 1 day, 1 week, 4 weeks and 12 weeks [13]. Scores were calculated as follows:

0: tissue defect or necrosis.

1: ulcer.

2: incomplete or partial epithelization.

3: complete epithelization.

The Dummett–Gupta Oral Pigmentation Index (DOPI) index was recorded to compare preoperative and postoperative measurement [24]. It was marked with 4 levels as follows:

DOPI score 0: no clinical pigmentation (pink-colored gingiva).

DOPI score 1: slight hyperpigmentation, light brown tissue.

DOPI score 2: moderate hyperpigmentation, medium brown tissue or a blend of pink and brown.

DOPI score 3: severe hyperpigmentation, dark brown or green/black tissue.

### 2.4. Statistical Analysis

Descriptive statistics were summarized for categorical (proportions and frequencies) and continuous variables (means, standard deviations). Statistical analysis was conducted by using SPSS 22.0 software for Windows (IBM Corp., New York, NY, USA), and Wilcoxon Signed Ranks Test was used to analyze DOPI index at baseline and 4 weeks posttreatment. A *p*-value ≤ 0.05 was considered statistically significant.

### 2.5. Ethics Approval

Ethical clearance for the study was obtained from the Ethics Committee of Can Tho University of Medicine and Pharmacy (approval No. 025/PCT–HĐĐĐ dated 15 December 2019). The purpose and procedures of the study were clearly given to patients and their volunteer participation was agreed by written informed consent.

## 3. Results

Among the studied subjects, 50.0% of participants were males and 50.0% were females. The average age was 22.75 age years (standard deviation = 0.93), with 97.4% in the 20–24 years age range.

Removal of the gingival pigmentation was performed without any bleeding. No carbonization happened in the process of laser treatment. No repetition of the hyperpigmentation method was performed during the postoperative interval. The wound healed completely in 4 weeks without any scar, infection or complication, which resulted in a significant improvement in esthetic appearance.

### 3.1. Location and Extent of Gingival Pigmentation

In the study, 36.8% had pigmentation in attached gingiva; 34.2% in attached, marginal and interdental gingiva; 18.4% in attached and marginal gingiva; 7.9% in marginal gingiva; and 2.7% in interdental gingiva. There were no cases with pigmentation in marginal gingiva and interdental gingival (Table 1).

This study presented all four degrees of gingival pigmentation as in Hedin’s classification. Out of 38 patients, two patients (5.2%) had one or two solitary unit(s) of pigmentation without the formation of a continuous ribbon; five patients (13.1%) had more than three units of pigmentation without the formation of continuous ribbon; nine patients (23.7%) had one or more short continuous ribbons of pigmentation; and 22 patients (57.8%) had one continuous ribbon (Table 2).

### 3.2. Pain after Treatment

As shown in Table 3, no one had severe pain after the treatment procedure. In total, 30/38 patients had pain after 1 day: mostly mild pain (29 patients), followed by moderate pain (1 patient). After 1 week, there was only one patient with mild pain left; after treatment at weeks 4 and 12, there was no pain in any patients.

### 3.3. Healing after Treatment

The result in Table 4 shows that all 38 cases had partial epithelization 1 day after treatment and complete wound healing was achieved after 1 week. The procedure of wound healing of a patient with gingival pigmentation was shown 7 days after treatment in this study (Figure 1).

### 3.4. Gingival Pigmentation According to DOPI after Treatment

Out of eight patients with a DOPI score of 1 before treatment, seven patients (87.5%) had a DOPI score of 0, while one patient (12.5%) still had a DOPI score of 1 after treatment. Out of 13 patients with a DOPI score of 2, three patients (23.1%) had a DOPI score of 1 and the remaining 10 patients (76.9%) had a DOPI score 0 at 1 week, 4 weeks and 12 weeks posttreatment. Of 17 patients with a DOPI score of 3, thirteen patients (76%) had a DOPI score of 1 and four patients (24%) had a DOPI score of 0. The changes in DOPI indices from baseline to 3 months were 55.6% for CO2-laser-treated sites (Figure 2 and Figure 3). The DOPI index after 4 weeks was significantly decreased compared with the baseline value (*p* < 0.001) (Table 5).

## 4. Discussion

In the present study, gingival hyperpigmentation in the attached gingiva had the highest number of cases and no melanin pigmentation in marginal gingiva and interdental gingiva was observed, unlike in the previous study of Ponnaiyan et al. [5] who reported that the majority (25.4%) of gingival pigmentation was in the attached gingiva and interdental gingiva, while there was 12.7% in marginal gingiva and interdental gingiva. The results of the studies were contrasting due to the different study designs and small sample sizes.

The rate of solitary and continuous pigmentation in the present study was lower than in the study of Hanioka et al. (2005) [25], who classified gingival pigmentation according to the presence of solitary pigmentation and the formation of continuous ribbons and reported that 29% to 32% patients had solitary units of pigmentation and 42% to 46% patients had continuous ribbons. However, the Hedin melanin index used to measure the extent of pigmented areas in our study was more specific than the classification used in Hanioka’s study.

A saline swab was used to scrub and remove the chromosome epithelium in the applied gingival areas and then a few small bleeding spots were found, but after the treatment, bleeding was no longer seen. A similar study was conducted by Hedge et al. (2013) [24], where they used Er: YAG laser with a 2.940 nm wavelength, 180 mJ energy, 10 Hz, 1.8 W power, long pulses and noncontact; the laser beam was guided in a “brushstroke” pattern to remove the entire pigmented gingiva. Some bleeding points were observed during the operation, and then no bleeding was found [26]. After lasing, swelling was not found because the CO_2_ laser causes prompt evaporation of the fluid in the cells, then destroys the cellular structure and may not release inflammatory chemical mediators. In addition to the evaporation of extracellular fluid, the denaturation of the protein structure occurs. A thin modified collagen layer on the surface of the laser-illuminated area can reduce the irritability of the tissue from causes in the oral cavity as an impermeable area as soon as the laser is introduced. Moreover, coagulation and the sealing of blood vessels 0.5 mm in diameter stop the bleeding [24].

The study conducted by Raaman et al. (2016) [27] reported that 6/25 patients had no pain on the first postoperative day, 10/25 patients had pain on the fourth day and no patient had pain on the seventh day after treatment, which is similar to the present study where 8/38 patients had no pain on the first day of treatment and only one patient had mild pain 1 week after treatment. In the recent study, lignocaine spray was applied to the operating area instead of local infiltration anesthesia, so postoperative pain was controlled better and the complete procedure was not interrupted. It is also consistent with the study of Hedge et al. (2013) [24] who observed that patients had no pain for 24 h and 1 week after treatment, whereas the study of Kishore et al. (2014) [13] showed that patients with Er:YAG laser treatment had mild pain, patients treated by CO_2_ laser had mild to moderate pain, and two patients had severe pain. Less pain after using the Er:YAG laser (2940 nm, 180 mJ, 10 Hz, total power of 1.8 W, long pulse) was due to the nerve endings sealing with the dense dry protein network in the gingival surface acting as a biofilm and the heat generation of the Er:YAG laser was shallower than the CO_2_ laser [28]. 

In this study, all patients had partial epithelization on post-operative day 1 (Figure 1b) and had complete epithelization the 1st week after treatment (Figure 1d). Hegazy et al. (2015) [16] used a CO_2_ laser at 3 W average power level and reported a similar result. The study conducted by Nagpal et al. (2015) [29] applied a diode laser with an 810 nm wavelength, 1 W power and continuous mode to treat gingival pigmentation in the first quadrant and epithelization was complete after 3 months of treatment, which is similar to the present study. Additionally, similar results were found by Ali et al. (2015) [7] using a diode laser with a wavelength of 810 nm at 0.5–1.5 W power in continuous mode and full epithelization was reported 2 weeks after the completion of treatment, while gingival tissues treated applying laser Er:YAG healed completely after 4 weeks, which was reported in the study of Pavlic et al. (2018) [14].

The present study revealed that 17 patients with a DOPI score of 3 had mild pigmentation (DOPI score 1) 4 weeks after treatment. It was consistent with the study conducted by Sukumar (2015) [30], in which one patient had a DOPI score of 0, eight patients had a DOPI score of 1 and one patient had a DOPI score of 2. After the 1st week, there was exfoliation of fibrin layer and re-epithelization, and the gingiva was healthy with no infection, swelling or scarring. After the 2nd week, gingival epitheliums were nonkeratinized and reddish gingiva was obtained. After the 4th week, the treated gingiva looked normal [31,32]. 

The relapse of pigmentation in interdental papilla and a few in attached gingiva was observed in the present study, which is similar to the study of Kishore et al. (2014) [13]. Moreover, Hegde et al. (2013) [24] reported that the relapse rate after 6 months of using CO_2_ laser for gingival hyperpigmentation was 22.8%, and the time to recurrence was 3 months after treatment.

The current study recognized that the use of a CO_2_ laser has advantages over gingival melanin hyperpigmentation treatment, such as hemostasis and less gingival deformity in gingival margin or interdental papilla. Nevertheless, the limitation of a CO_2_ laser is that it cannot entirely remove melanin pigment in deep layers due to the effect on the bone and tooth structure. Additionally, the overactivity, proliferation and transfer of residual pigment cells after performing the CO_2_ laser procedure explains the repigmentation in our study [28].

## 5. Conclusions

According to the results of the study, using a super-pulsed CO_2_ laser was a safe and effective modality for removing melanin pigmented gingiva. Patients’ pain experience was reduced after 1 week of treatment and a better healing process was shown at 7 days post-operatively. Moreover, the aesthetic outcome after CO_2_ laser-assisted treatment of pigmentation was improved. Therefore, it is clear that the efficacy of the CO_2_ laser method is obtained, and it is recommended that clinicians consider the CO_2_ laser method as an option for pigmented gingiva removal for their patients’ treatment.

## Figures and Tables

**Figure 1 dentistry-10-00238-f001:**
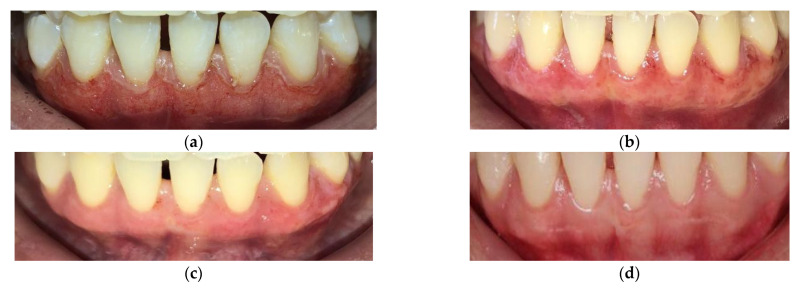
The wound healing process in the anterior mandibular gingiva after the depigmentation treatment. (**a**) Immediately after lasing; (**b**) after 1 day—score 2; (**c**) after 4 days—score 2; (**d**) after 7 days—score 3.

**Figure 2 dentistry-10-00238-f002:**
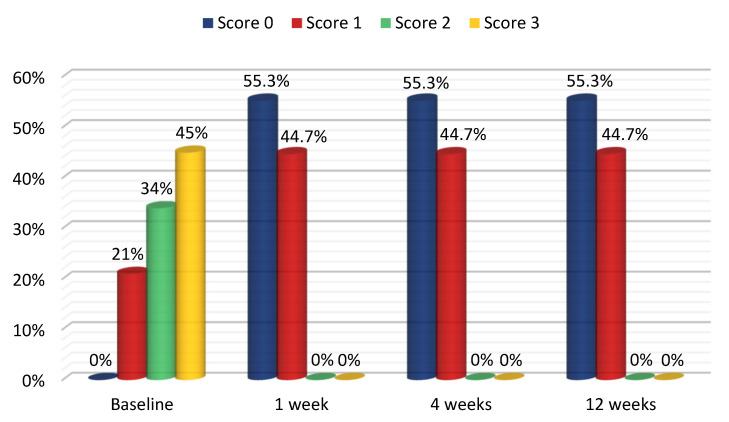
DOPI score using CO_2_ laser at weeks 1, 4 and 12 after treatment.

**Figure 3 dentistry-10-00238-f003:**
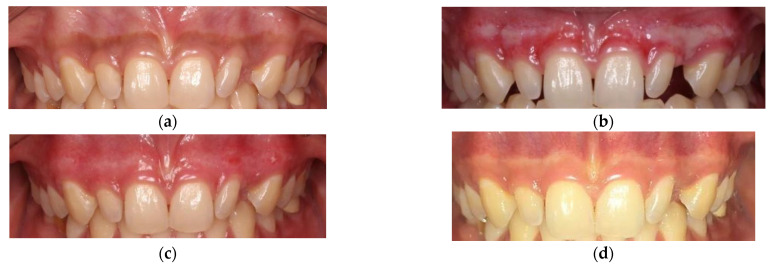
Gingival depigmentation at 12-week follow-up. (**a**) Before treatment—DOPI score 3; (**b**) 1 week treatment—DOPI score 1; (**c**) 4 weeks treatment—DOPI score 1; (**d**) 12 weeks treatment—DOPI score 1.

**Table 1 dentistry-10-00238-t001:** Location of gingival pigmentation (Ponnaiyan’s classification).

Variable	Frequency		
	n (%)	n (%)	n (%)
	Maxillary	Mandibular	Total
Class I	5 (13.1)	9 (23.7)	14 (36.8)
Class II	4 (10.5)	3 (7.9)	7 (18.4)
Class III	9 (23.6)	4 (10.6)	13 (34.2)
Class IV	2 (5.2)	1 (2.7)	3 (7.9)
Class V	1 (2.7)	0 (0)	1 (2.7)
Class VI	0 (0)	0 (0)	0 (0)

**Table 2 dentistry-10-00238-t002:** Extent of gingival pigmentation (Hedin’s classification).

Variable	Frequency		
	n (%)	n (%)	n (%)
	Maxillary	Mandibular	Total
Degree 1	0 (0)	2 (5.2)	2 (5.2)
Degree 2	1 (2.7)	4 (10.5)	5 (13.1)
Degree 3	6 (15.8)	3 (7.9)	9 (23.7)
Degree 4	13 (34.1)	9 (23.7)	22 (57.8)

**Table 3 dentistry-10-00238-t003:** Pain at 1 day, weeks 1, 4 and 12 after treatment.

Variable Pain	Time			
	n (%)	n (%)	n (%)	n (%)
	1 Day	1 Week	4 Weeks	12 Weeks
No	8 (21)	37 (97)	38 (100)	38 (100)
Mild	29 (76)	1 (2.6)	0 (0)	0 (0)
Moderate	1 (2.6)	0 (0)	0 (0)	0 (0)
Severe	0 (0)	0 (0)	0 (0)	0 (0)

**Table 4 dentistry-10-00238-t004:** Healing at day 1, week 1 and 4 after treatment.

Variable Healing	Time		
	n (%)	n (%)	n (%)
	1 Day	1 Week	4 Weeks
Score 1	0 (0)	0 (0)	0 (0)
Score 2	0 (0)	0 (0)	0 (0)
Score 3	38 (100)	0 (0)	0 (0)
Score 4	0 (0)	38 (100)	38 (100)

**Table 5 dentistry-10-00238-t005:** Pre- and 4-week postoperative comparison of DOPI index.

Variable	Baseline	4 Weeks
Mean ± SD	2.24 ± 0.79	0.97 ± 0.16
*p* *	<0.001	

* Wilcoxon Signed Ranks Test, statistically significant *p*-value ≤ 0.05.

## Data Availability

The data of this study are available from the corresponding author on reasonable request via thtrung@ctump.edu.vn.

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
