# Peer review of "Evaluation of Carbon Dioxide Laser–Assisted Treatment for Gingival Melanin Hyperpigmentation"

_dentistry, 2022, doi:10.3390/dj10120238_

Round 1

Reviewer 1 Report

Thank you for the work. I m generally happy to read the content of the current study. However, please see my suggestions to improve your work.

Abstract 

Explanation for abbreviation CO2 laser was missing in all text.

Mat and Methods

Detail information about stat analysis is missing? Did the authors calculate the sample size and any power analysis done in the current study?

ALL materials and methodologies used, please use general names with your text, followed by (brand names; manufacturer, city, country) in parentheses.

Pls clearly state the mode of CO2 laser used in the current study?

Results

Caption for Figure 1 must be revised (it should be the scoring used in the current study)?

Figure 2 must be revised.

Score 3 after 12 weeks of treatment 

The authors mentioned the % of number of participants after tx at different time points, but figure 2 show the % of scoring after tx at different time points, meaning that figure and the results are not related so my opinion is the results should be rewritten in that part. 

Discussion

Pls clearly describe why the authors used low energy levels in CO2 laser surgery?

“The post–operative swelling was not present because a wound caused by using CO2 laser is not a burn, it causes almost immediate evaporation of the fluid in the cells, so destroys the cellular structure, whereas may not release inflammatory chemical mediators.”

Cmt – unclear sentence.

“The study done by Raaman et al. (2016) [22] presented that 6/25 patients had no pain on 1st post–operative day, 10/25 patients had pain on 4th day and no patient had pain on 7th day after treatment, which is similar to the present study where 8/38 patients had no pain on 1st day of treatment and only one patients had mild pain after 1 week treatment. In the study, lignocaine spray was applied to the operating area instead of local infiltration anesthesia to achieve better post–operative pain control. It is also consistent with the study of Hedge et al. (2013) [19] observed that patients were no pain for 24 hours and 1 week after treatment, whereas, Kishore et al. (2014) [10] compared two Er:YAG and CO2 laser methods on each group of 10 patients and concluded that patients in group of Er:YAG laser had mild pain, while patients in group of CO2 laser had mild to moderate pain, and 2 patients with severe pain. Less pain after using the Er:YAG laser (2940 nm, 180 mJ, 10 Hz, total power of 1.8 W, long pulse) is due to nerve endings seal of a dense dry protein network in gingival surface as a biofilm. Also, Er:YAG laser has lower energy level and 1 μm beam depth, which is less or painless than CO2 lasers [23].”

Cmt – unclear information, should be rewritten.

Conclusion 

It should be rewritten and should be focused on the current study results (should not be overgeneralized)

References

Double-checked ref no 12 and 17.

General comments

1) Title should be revised?

2) It is strongly recommended that the revised manuscript is reviewed to correct any grammatical or spelling errors

Round 2

Reviewer 1 Report

Thank you for considering my comments. 

Author Response

Thank you very much. We sincerely appreciate all your valuable comments and suggestions.

Reviewer 2 Report

Dear Author

Thank you for your corrections, I recommend in the results section to highlight (p-value) of the comparison preoperative data of the pigmented area with that of postoperative data after 4 weeks from treatment day.

Best regards

Author Response

Dear Reviewer,

Thank you for your recommendation.

In the latest version, we have updated that Wilcoxon Signed Ranks Test was used to analyze DOPI index at baseline and 4 weeks post–treatment and highlighted the p-value in Table 5 of the results section.

"Table 5. Pre– and 4 weeks post–operative comparison of DOPI index.

Variable

Baseline

4 weeks

Mean ± SD

2.24 ± 0.79

0.97 ± 0.16

p*

< 0.001

* Wilcoxon Signed Ranks Test, statistically significant p–value ≤ 0.05."

Yours sincerely